# Spectral–Spatial Feature Fusion for Hyperspectral Anomaly Detection

**DOI:** 10.3390/s24051652

**Published:** 2024-03-03

**Authors:** Shaocong Liu, Zhen Li, Guangyuan Wang, Xianfei Qiu, Tinghao Liu, Jing Cao, Donghui Zhang

**Affiliations:** Institute of Remote Sensing Satellite, China Academy of Space Technology (CAST), Beijing 100094, China

**Keywords:** hyperspectral image, isolation forest, local saliency detection, anomaly detection, spectral–spatial fusion

## Abstract

Hyperspectral anomaly detection is used to recognize unusual patterns or anomalies in hyperspectral data. Currently, many spectral–spatial detection methods have been proposed with a cascaded manner; however, they often neglect the complementary characteristics between the spectral and spatial dimensions, which easily leads to yield high false alarm rate. To alleviate this issue, a spectral–spatial information fusion (SSIF) method is designed for hyperspectral anomaly detection. First, an isolation forest is exploited to obtain spectral anomaly map, in which the object-level feature is constructed with an entropy rate segmentation algorithm. Then, a local spatial saliency detection scheme is proposed to produce the spatial anomaly result. Finally, the spectral and spatial anomaly scores are integrated together followed by a domain transform recursive filtering to generate the final detection result. Experiments on five hyperspectral datasets covering ocean and airport scenes prove that the proposed SSIF produces superior detection results over other state-of-the-art detection techniques.

## 1. Introduction

A hyperspectral image (HSI) contains hundreds of narrow spectral channels for each pixel, which delivers rich spectral and spatial information [1,2,3,4,5]. Owing to this virtue, HSI has been popularly employed in a considerable number of fields, such as object detection [6,7,8,9], image classification [10,11,12,13], and change detection [14,15,16]. For all the application fields, hyperspectral anomaly detection, which aims to recognize the outliers whose spectra are significantly different from an ambient scene, has drawn much more attention over the last few years [17].

In the last several decades, all kinds of hyperspectral anomaly detection techniques have been developed, which can be loosely grouped into the following several types: statistical models, subspace-based approaches, reconstruction-based approaches, and deep learning approaches. The statistical models are the most typical hyperspectral anomaly detection techniques. A representative statistical technique is the Reed–Xiaoli (RX) scheme [18], which presumes that the multivariate Gaussian distributions can be exploited to model the background and predict the probability density functions of background samples. The RX method contains two versions, i.e., local RX and global RX [19]. When the whole image is used to model Gaussian background distribution, it is named as global RX. When the RX method gauges the Gaussian model using local image pixels, it is regarded as the local RX. Nevertheless, real-world HSIs have extremely complex background, and it is hard to characterize it with a multivariate Gaussian distribution. Thus, some improved versions have been studied, such as subspace RX, kernel RX, and linear filter-based RX. For example, Heesung et al. [20] developed a kernel RX detection technique, in which the hyperspectral cube was first projected into a nonlinear feature space, and then the multivariate normal distribution was employed to fit the feature data. Guo et al. [21] developed a linear filter-based RX detection method for hyperspectral images by decreasing the weight of anomalous objects or noisy instances and enhancing one of the background pixels.

The subspace-based methods postulate that the anomaly and background reflectances have high separability in a feature subspace [22,23,24]. The orthogonal subspace projection (OSP) is a classical subspace-based object detection technique. For instance, Chang et al. [25,26] applied an improved OSP technique for the anomaly detection of hyperspectral data, where the automatic object detection process was implemented on the background pixels. Xiang et al. [27] combined a local joint subspace process and classifier for hyperspectral anomaly detection. Chang et al. [28] developed a subspace selection-based isolation forest model for detecting anomaly objects in hyperspectral images, which used the subsampling technique rather than modeling the whole background. Furthermore, some improved detection methods have been also investigated, such as local 3D OSP [29], nonparametric OSP [30], and multiple subspaces [31].

The reconstruction-based methods aim to recover the input data with a certain model, and the residual represents the probability of a pixel belonging to anomaly or background. Representative reconstruction-based object detection methods mainly include low-rank representation [32,33], sparse representation (SR) [34,35], and collaborative representation [36,37]. For example, Zhu et al. [34] developed an adaptive weighted SR technique for detecting anomaly targets in hyperspectral data, in which a random selection strategy was employed to form the background dictionary. Li et al. [36] applied a collaborative representation for the anomaly detection of hyperspectral data, which assumes that the pixel reflectances belonging to the background area could be characterized by its adjacent pixels while the anomalous pixels cannot. Sun et al. [32] developed a low-rank and sparse matrix decomposition technique to detect the anomaly targets, where the background pixels were the low-rank part and the anomalies were thinly scattered in the whole data.

Recently, the deep learning networks have been also developed to identify the anomaly targets in hyperspectral data. Two typical unsupervised deep networks are utilized for hyperspectral anomaly detection, i.e., the generative adversarial network [38,39,40] and the autoencoder model [35,41,42]. For instance, Jiang et al. [38] applied a generative adversarial network for detecting the anomaly objects, which assumed that the amount of background pixels was significantly greater than one of target pixels. Lu et al. [41] designed a manifold constrained autoencoder for the anomaly detection of hyperspectral data, in which the manifold constraint was utilized to model the embedding representation and the autoencoder network was employed to pertain the latent intrinsic structures. Wang et al. developed a sliding dual-window-guided reconstruction network to detect the anomalies in HSIs, in which the outer window information was utilized to predict the pure background [43]. Ren et al. proposed a unified nonconvex framework with generalized shrinkage mappings to approximate the group sparsity, $l_{0}$ gradient, and low-rankness penalties in the low-rank-based anomaly detection methods [44]. Lin et al. designed a low-rank and sparse constrained deep autoencoder for hyperspectral anomaly detection [45]. In addition, some supervised deep learning networks have been also applied for detecting anomaly targets in hyperspectral data [46,47,48].

Although these approaches mentioned above use spectral and spatial information in a cascaded manner for hyperspectral anomaly detection, they ignore the complementary property between the spectral and spatial dimension. To alleviate this issue, several publications focus on the fusion of spectral–spatial information [49,50,51]. They use different spectral–spatial feature extractors and a simple concatenation way for hyperspectral anomaly detection. The complementary information between spectral and spatial branches is not considered, which tends to yield high false alarm rate. To remedy this issue, a spectral–spatial feature fusion method is proposed for hyperspectral anomaly detection, in which the isolation forest method is used to estimate the spectral anomaly score while the local spatial saliency detection method is utilized to obtain the spatial anomaly score. To be specific, first, a superpixel-level isolation forest method is constructed to obtain the spectral anomaly score. Then, a local spatial saliency detection method is introduced to yield a spatial anomaly score by using local spatial similarity in the background pixels. Finally, the spectral and spatial anomaly scores are integrated together, followed by an edge-preserving filter to generate the final detection result. Experiments on several real-world hyperspectral datasets reveal that the proposed SSIF can obtain ascendant detection results over other hyperspectral anomaly detection approaches. The key contributions of this work are concluded as follows:(1)A spectral–spatial feature fusion framework is proposed for hyperspectral anomaly detection, which makes full use of the object anomalies from both the spectral and spatial dimensions;(2)A superpixel-level isolation forest is designed to exploit the homogeneity of objects, which can preserve the object structures well;(3)Experiments on five hyperspectral datum reveal that the proposed SSIF can significantly decrease the false alarm rate compared to other detection techniques.

The remaining parts of this paper are grouped into the following five sections: Section 2 portrays the related work. Section 3 provides the detailed steps of the designed detection scheme. Section 4 analyzes the subjective and objective results of all studied methods. Section 5 provides the ablation experiments. Section 6 sums up the conclusions.

## 2. Related Work

### 2.1. Isolation Forest Algorithm

Isolation forest (iForest) was first designed for outlier detection [52]. Its core idea is that anomaly cases in a given data set are typically uncommon and distinct from regular instances, making them more prone to be separated in various binary tree architectures than the normal examples. Specifically, the input data are first separated by a randomly threshold, resulting in a tree construction from the root to the leaf. Every tree is further grown until every instance fits the predefined condition. Then, the path length, i.e., the isolation depth, of each instance is calculated with the amount of edges that the instance traverses a binary tree from the root node to the leaf node. As a result, the binary tree is recursively segmented, in which the anomalous instances arrive at the leaf nodes quickly, whereas the normal instances must undergo many more splits before they arrive. In general, the path length of the anomalous instances is shorter than one of the normal instances. Finally, the anomaly value is counted by averaging the path length across multiple binary trees.

Figure 1 depicts the principle of the isolation forest. As can be seen, the normal example $x_{b}$ needs more times to achieve segmentation, while the anomalous example $x_{a}$ is more easily isolated. Accordingly, the path length of $x_{b}$ is higher than one of $x_{a}$; therefore, the merit of the iForest is that the outliers can be identified according to the principle of the isolation tree without introducing any measurement, which can improve the computing efficiency with respect to density or distance-based detection algorithms.

### 2.2. Domain Transform Recursive Filtering

Domain transform recursive filtering (DTRF) is a representative edge-preserving filtering [53], which can detach the useless information well while preserving the image edges and structures. Due to this advantage, the DTRF has generally been utilized in numerous aspects, such as image interpretation, object identification, and image visualization. Assume I to be the input data, the transformed data F is computed as follows:(1)$$F_{i}=I_{0}+\sum_{j=1}^{J}\left(1+\frac{\delta_{s}}{\delta_{r}}|I_{j}-I_{j-1}|\right)$$
where $\delta_{s}$ and $\delta_{r}$ mean two free parameters, which are used to regulate the smoothness degree. Afterwards, the input data I is further processed with recursive filter.
(2)$$J_{i}=\left(1-a^{b}\right)I_{i}+a^{b}J_{i-1}$$
where $J_{i}$ indicates the filtered result of *i*th pixel. $a=\exp(-\sqrt{2}/\delta_{s}\;)$ stands for a feedback value. *b* stands for the distance between $F_{i-1}$ and $F_{i}$. In this work, the DTRF is denoted as $\mathrm{DTRF}(I,G,\delta_{s},\delta_{r})$, where I refers to the input data, and G denotes the guidance data that is utilized to the distance *b*.

## 3. Proposed Method

Figure 2 depicts the schematic of the proposed spectral–spatial information fusion detection framework, which is composed of three stages. First, the superpixel-level isolation forest is designed for calculating the spectral anomaly map. Then, the spatial similarity measurement is employed to calculate the spatial anomaly result. Finally, the spectral and spatial anomaly scores are merged together and the DTRF is leveraged to optimize the fused map so as to generate the ultimate detection map.

### 3.1. Spectral Anomaly Detection

A hyperspectral image contains abundant spectral information of ground targets. The subtle spectral information provides an unique diagnostic capability for object detection. To take full advantage of the spectral information in HSIs, an object-level isolation forest is developed for spectral anomaly detection in this work, which consists of the following three steps: superpixel segmentation, isolation tree construction, and anomaly score estimation.

(1) Superpixel segmentation: Assume I to be the input hyperspectral image, a principal component analysis technique is first conducted on the input data I to obtain the base image B, where the fist principal components are viewed as the base image. Next, the entropy rate superpixel (ERS) segmentation scheme [54] is introduced to segment the base image B.
(3)P=ERS(B,K)
where P denotes the segmentation map. ERS represents the ERS algorithm. *K* is the amount of superpixels. Here, *K* is calculated as follows:(4)$$K=\left\lfloor N\times \frac{\tilde{M}}{M}\right\rfloor$$
where *N* is an predefined parameter. $\tilde{M}$ denotes the proportion of nonzero elements in the binary detected map that is measured by using the Sobel filter on the base image B. *M* stands for the amount of all pixels presented in the whole image. According to the segmentation result of the B, the position location of pixels for each superpixel $P_{i}$ can be obtained. Accordingly, the 3D superpixels $X_{i},i=1,2,\cdots ,K$ in HSI can be easily attained.

(2) Isolation forest construction: The obtained superpixel HSI is utilized to construct an isolation forest. First, *T* pixels are initially stochastically chosen from the input data $X_{i}$. Then, the chosen pixels are grouped into a left node and a right node according to a simple decision strategy. In more detail, if $X_{s}^{t}$ is lower than the split value η, the *t*-th pixel is grouped into the left node. On the contrary, it is grouped into the right node. Here, *s* denotes a random number from 1 to *D*. The split threshold η is randomly chosen between the minimum and maximum of X. Next, each child node is further segmented with the same operation above iteratively, until one of the following nether conditions is required: (1) the pixels in each child node are consistent; (2) the amount of pixels in each child node is 1; (3) the number of tree comes up to the maximum height $H_{max}$. In this paper, $H_{max}=log_{2}T$. At last, the procedure steps of the isolation tree are iterated *q* times to form the isolation forest.

(3) Anomaly score estimation: The established isolation forest is exploited to gauge each pixel’s anomaly score. Specifically, the amount of edges is considered as the path length of each pixel for every isolation tree, in which the pixel varies from the root to the terminal node. Since anomaly objects typically have relatively small areas and distinctive spectrum compared to surrounding background, they are readily segregated from the external nodes. Accordingly, their path lengths are shorter, while the background pixels have longer paths. Based on this observation, the path length is utilized to estimate anomalies. It is important to mention that the path length of individual pixel is dissimilar to all established trees. Thus, the eventual path length can be calculated with the averaging operation on all isolation trees.

The iForest has *q* isolation trees $\left\{Q_{1},Q_{2},...,Q_{q}\right\}$. For a test pixel x∈X, $h_{i}\left(x\right)$ denotes the path length of *x* in $Q_{i}$, the average path length on the whole isolation trees is computed as follows:(5)$$A\left(h\left(x\right)\right)=\frac{1}{T}\sum_{i}^{T}h_{i}\left(x\right)$$
Then, for test pixel *x*, we can compute its anomaly value s∈(0,1] as follows:(6)$$s_{1}\left(x\right)=2^{-\frac{A(h(x))}{c(M)}}$$
Here, c(m)=2H(m−1)−(2(m−1)/m). H(m) is set as ln(m)+0.5772156649 (Euler’s constant). When the above operation is performed on each superpixel, the anomaly map $S_{1}$ can be estimated.

### 3.2. Spatial Anomaly Detection

The spatial shape of the anomaly object is different from its adjacent areas [55]. Based on this observation, a local saliency measurement scheme is exploited to estimate the spatial anomaly score, which computes the spatial difference between the center pixel and its ambient background.
(7)$$s_{2}=\min\left(tr{\left(C_{w}-C_{w_{i}}\right)}^{T}\cdot tr\left(C_{w}-C_{w_{i}}\right)\right),i=1,2,..,W$$
where $s_{2}$ is the spatial anomaly score. $C_{w}$ indicates the center window. $C_{w_{i}}$ represents the surrounding background window. *W* is the window size. Figure 3 presents an illustration of local saliency detection. By sliding the red window around the blue window, the local difference information between the center window and its surrounding background windows can be calculated. The higher the value, the more likely it is to be the background.

### 3.3. Decision Fusion

To fully utilize the complementary characteristics between the spatial and spectral domains, the spectral and spatial scores are first merged together.
(8)$$R=S_{1}\times S_{2}$$
Then, the fused detection map is further optimized with the DTRF so as to remove the noisy pixels.
(9)$$\tilde{R}=\mathrm{DTRF}\left(R,B,\sigma_{s},\sigma_{r}\right)$$
where $\tilde{R}$ represents the final detection map. DTRF is the DTRF. The base image B is considered as the guided image. $\sigma_{s}$ and $\sigma_{r}$ are fixed as 5 and 0.5, respectively.Algorithm 1 presents the pseudocode of the proposed method.
**Algorithm 1** Spectral–spatial feature fusion for hyperspectral anomaly detection**Input:**      Input hyperspectral data I;**Output:**      Detection result $\tilde{R}$   1:According to (Equation 3), calculate the 2D superpixel segmentation result P   2:Calculate the corresponding 3D superpixels X in HSI according to the coordinate of pixels in each 2D superpixel $P_{i}$.   3:Construct the isolation forest based on the 3D superpixel in HSI.   4:Combine the constructed forest and Equation (Equation 6) to compute the spectral anomaly score $S_{1}$.   5:According to Equation (Equation 7), compute the spatial anomaly score $S_{2}$.   6:According to Equation (Equation 8), fuse the spectral and spatial anomaly scores to generate the fused detection result R.   7:According to (Equation 9), integrate the spectral and spatial anomaly results to generate the final detection result $\tilde{R}$  8:**Return **$\tilde{R}$

## 4. Results

In this part, several experiments are designed to validate the superiority of the proposed SSIF. The recorded results are listed in the subsequent parts. All experiments are implemented on a laptop, which is equipped with 64GB RAM and Core i9-10900K CPU.

### 4.1. Experimental Setup

(1) Datasets: In this work, five real hyperspectral datasets from different imaging scenarios, i.e., Beach, Pavia city, San Diego-I, San Diego-II, and Gulfport, are exploited to judge the detection effect of all the considered techniques. Table 1 shows the detailed information of the used datasets.

The Beach data were gained by the Airborne Visible/Infrared Imaging Spectrometer (AVIRIS) device over Cat Island. These data embody 224 reflectance channels varying from 0.4 to 2.5 μm. Before the experiments, several water absorption and noisy channels are removed, and 188 channels are preserved. Its size is of 150 × 150, and each pixel accounts for 17.2 m. Figure 4 presents the pseudocolor image and reference map.

The Pavia city data were collected by the Reflective Optics System Imaging Spectrometer (ROSIS) equipment. It is about Pavia city scene. These data are of 150 × 150 pixels with a spatial resolution 1.3 m. The imaging range is from 0.43 to 0.86 μm with 205 spectral bands. This scene has water, a bridge , bare soil, and buildings. The pseudocolor RGB and reference image are depicted in Figure 5.

The San Diego-I dataset was acquired by the AVIRIS device over the airport region of San Diego, USA. These data’s spatial size are 100 × 100 pixels, and each pixel accounts for 3.5 m. The imaging scope varies from 0.4 to 2.5 μm with spectral interval of 10 nm. After discarding water absorption and noisy channels, 189 channels are employed for experiments. Figure 6 gives the pseudocolor image and reference image.

The San Diego-II dataset is situated at the heart of the airport area of San Diego, which is named San Diego-II. The spatial size of these data are 100 × 100. This scene is composed of exposed soil, hangers, parking aprons, and airports, in which three airports are taken as anomalies. Figure 7 gives the pseudocolor image and ground truth.

The Gulfport data were obtained by the AVIRIS device about the airport region of Gulfport, USA. This image contains 191 spectral channels varying from 0.55 to 1.85 μm. Its spatial size is of 100 × 100 pixels, and each pixel accounts for 3.4 m. These data includes a highway, an airport runway, vegetation, and an airport. The airport is considered as an anomaly. Figure 8 depicts the sample image and reference image.

(2) Objective Indexes: To quantitatively assess the detection effect of different methods, two widely used metrics are adopted, i.e., the receiver operating characteristic (ROC) curve and the area under the curve (AUC). The ROC curve provides the correlation between the detection probability (DP) and the false alarm rate (FAR) at different threshold settings. In more detail, when the detection result and the reference image are given, the DP and FAR are calculated as follows:(10)$$\mathrm{DP}=\frac{N_{D}}{N_{T}}\;\mathrm{FAR}=\frac{N_{F}}{N}$$
where $N_{D}$ means the amount of identified object pixels below a fixed threshold. $N_{T}$ means the total amount of true object region in the original dataset. $N_{F}$ indicates the sum of false alarm pixels. *N* indicates the sum of pixels in the input image. When the DP value is higher than one of other approaches at the same false alarm, it means that this method with a higher DP yields the best detection result.

The AUC value is estimated with the whole region under the ROC curve. A higher AUC represents a better anomaly detection result. The AUC is given by:(11)$$\mathrm{AUC}=\int_{-\infty }^{+\infty }DP(H)FAR(H)dH$$
where DP(H) denotes how many correct positive samples occur among all examples when the threshold is *H*. FAR(H) records how many incorrect positive samples occur among all negative examples.

### 4.2. Detection Results

To illustrate the strength of the proposed SSIF, several hyperspectral anomaly detection techniques are selected as compared techniques, including RX [18], CRD [36], AED [56], KIFD [57], PTA [58], RGAE [59], and SSIIF [60]. These approaches are adopted since they are representative or recently developed techniques. Specifically, the RX technique is a classical statistical modeling technique. The CRD is a popular collaborative representation scheme. The AED is based on attribute filters. The PTA is tensor representation technique. The KIFD and SSIIF are the representative iForest methods. The RGAE is based on the graph autoencoder model. The related parameters in these detection methods follow the corresponding publication.

Figure 9, Figure 10, Figure 11, Figure 12 and Figure 13 depict the detection results of all techniques on five datasets. It can be found that the RX scheme yields unsatisfactory detection maps. Many anomalies cannot be detected well in the resulting maps. For example, the RX method fails to identify three airplanes from the San Diego-I dataset. For the CRD scheme, the abnormal objects are hardly detected. The background information is regarded as the anomalies. For the AED method, although the anomalies can be effectively detected, the similar land covers in the background also are viewed as anomalies. For example, the island in the Beach dataset is mistakenly detected. The KIFD and PTA techniques can detect the locations and shapes of different objects well; however, they obtain very high false alarm rates for all datasets. The reason is that the KIFD and PTA methods fail to remove the interference of the background information. The RGAE method suffers from poor detection performance. The airplanes in the San Diego-I, San Diego-II, and Gulfport datasets cannot be effectively detected. The SSIIF method slightly improves the detection performance; nevertheless, the background information still can be found in the detection results. By contrast, the proposed SSIF yields the best detection maps with the lowest false alarm rate among all the considered approaches. The background information can be well-suppressed, and only anomalies are effectively separated from the original image.

Furthermore, the objective detection performances of the different approaches are quantitatively assessed via the AUC score. Table 2 lists the AUC scores of all the studied detection approaches, and the best detection score in each row is displayed in bold. It is obvious that the proposed SSIF produces the highest AUCs on all images. This also illustrates that the proposed method indeed yields the best detection effect compared to other techniques, as the RGAE method cannot achieve stable detection results. For the San Diego-I and Gulfport datasets, the RGAE method yields the lowest AUCs; moreover, the RX and CRD methods also obtain a relatively low detection accuracy. Generally, among all the studied detection techniques, the proposed SSIF achieves the highest objective accuracy and produces satisfactory detection results.

Figure 14 depicts the ROC curves of all the detection approaches on all images. It can be observed that the detection probability of the proposed SSIF is always greater than the other compared techniques when the FAR is varying from 0 to 1. This proclaims that the proposed SSIF yields superior detection results among all the detection approaches. For the San Diego-II dataset, the proposed SSIF has a slightly lower probability of detection than the RGAE method when the FAR is from 0.05 to 0.1. In general, by observing the ROC curves of Figure 14, it is obvious that the designed SSIF is higher than other approaches.

## 5. Discussion

### 5.1. The Influence of Degradation Factors

To illustrate the influence of different degradation factors on anomaly detection, an experiment is performed on the Gulfport dataset, in which the input dataset is contaminated with different factors, including Gaussian noise with a variance of 0.1 , shadow, and fog. It should be noted that all the degradation models are from reference [61]. Figure 15 shows the detection results of the proposed method under different degradation factors. It is shown that when the source image is polluted with shadow, the detection accuracy of the proposed method is significantly decreased, i.e., the AUC is decreased by 14.31%. Among all the degradation factors, shadow has a negative impact on target detection performance.

### 5.2. The Influence of Different Parameters

In this subsection, the influence of all the free parameters in the spectral anomaly detection branch, i.e., the amount of trees *q* and the amount of superpixels *N*, to the detection effect of the proposed method is discussed. Figure 16a depicts the influence of different numbers of isolation trees. It can be found that when the amount of isolation trees is relatively small, the proposed SSIF produces a low detection accuracy. When the amount of isolation trees is greater than 50, the detection effect is satisfactory. Furthermore, as there is an increase in the amount of isolation trees, the computing cost of the proposed framework has an increasing trend. In this work, the amount of isolation trees *q* is set as 50. Besides, Figure 16b presents the AUCs of the proposed SSIF with different numbers of superpixels. It is shown that when the number of superpixels is relatively large, the detection accuracy tends to decrease. The reason is that excessive superpixels leads to an image oversegmentation issue. In addition, a quantity of superpixels causes each local region containing fewer pixels, resulting in unstable detection results. In this work, the number of superpixels *N* is set as nine for all experiments.

The window size *W* in the spatial anomaly detection branch also needs to be determined. Figure 17 depicts the detection accuracy of different sizes of windows in the spatial anomaly detection branch. The window size *W* varies from 1 to 13 with step size 2. When the window size *W* increases, the AUC of the proposed SSIF tends to decrease. This is because useless spatial information is used to measure the difference between the center window and the local window. By observing Figure 17, it is obvious that when the window size *W* is three, the detection effect of the proposed method is the highest; thus, the window size *W* is set as three for the following experiments.

### 5.3. The Influence of Different Components

In this part, the influence of different components in the proposed framework on the detection performance is analyzed. Table 3 lists the quantitative scores of different components in the developed framework. By comparing the spatial branch and spectral branch, it is shown that the spectral branch yields a lower detection accuracy. The reason is that hyperspectral image usually suffers from spectral mixture issue. By fusing the spectral and spatial branches, the detection accuracy gains great improvement. For example, the AUC value of the proposed SSIF is increased by 1.92% more than the one of the spectral branch. This experiment also illustrates that the spectral and spatial information fusion can boost the detection effect well.

### 5.4. Computing Time

In this part, the running time of different detection schemes on all datasets is discussed. Table 4 presents the computational cost of all detection schemes. It is shown that the RX detector performs the fastest. The reason is that this method only requires a simple distance calculation; however, the RX method produces a relatively poor detection result. The computational burden of the RGAE scheme is the highest among all the studied techniques since the autoencoders require a plentiful amount of iterations during the training process. By contrast, the running time of the proposed SSIF is quite competitive among all the studied techniques. Furthermore, as the spatial size increases, the computational cost tends to increase. How to decrease the computing time is still an interesting topic.

## 6. Conclusions

In this study, a spectral–spatial feature fusion is designed for hyperspectral anomaly detection, which is comprised of three main stages. First, an object-level isolation forest is constructed to estimate the spectral anomaly score. Then, a local spatial similarity strategy is exploited to produce the spatial anomaly score. Finally, the spectral and spatial anomaly probability maps are merged together, followed by a edge-preserving filtering to yield the ultimate detection map. Experiments on five real-world hyperspectral datasets covering ocean and airport scenes verify that the proposed SSIF can obtain better detection results with respect to other hyperspectral anomaly detection approaches. In addition, the proposed SSIF can suppress the background information well, and achieves a substantially lower false alarm rate. It should be mentioned that when the proposed method is applied in real anomaly detection, the input data needs to undergo atmospheric correction. In the future, we will focus on developing real-time hyperspectral anomaly detection techniques; furthermore, we will expand the applications of hyperspectral anomaly detection. For example, we can combine it with point target detection so as to achieve infrared dim small flying target recognition [62].

## Figures and Tables

**Figure 1 sensors-24-01652-f001:**
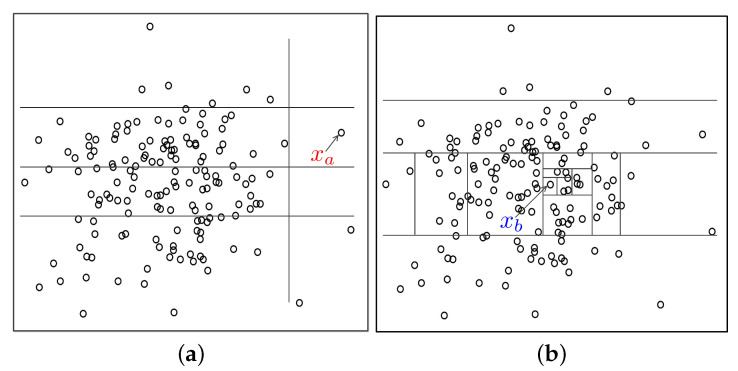
Principle of the isolation forest. (**a**) Anomalous example is isolated with four segmentations. (**b**) Normal example is isolated with 13 segmentations.

**Figure 2 sensors-24-01652-f002:**
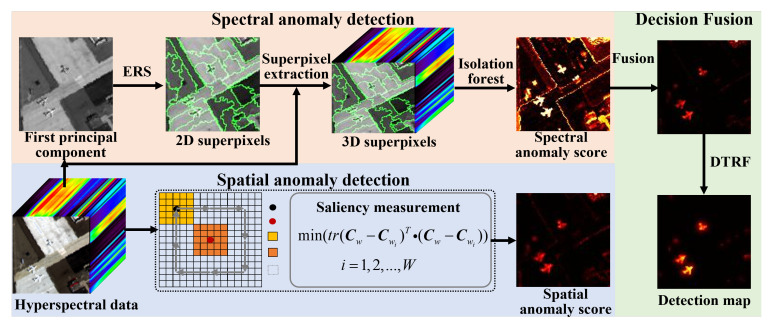
The schematic of the SSIF detection method.

**Figure 3 sensors-24-01652-f003:**
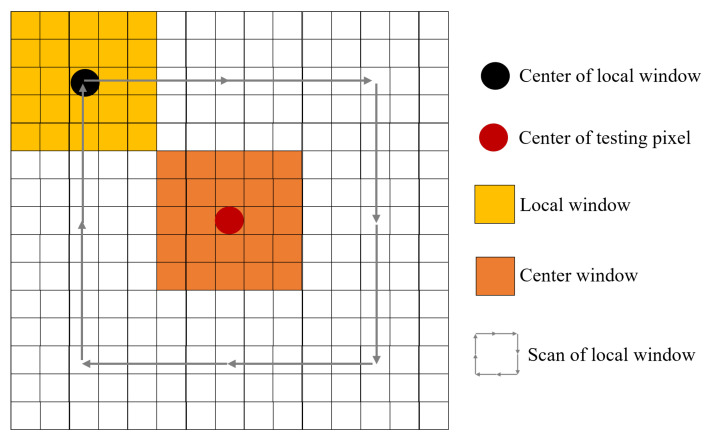
An illustration of local saliency detection.

**Figure 4 sensors-24-01652-f004:**
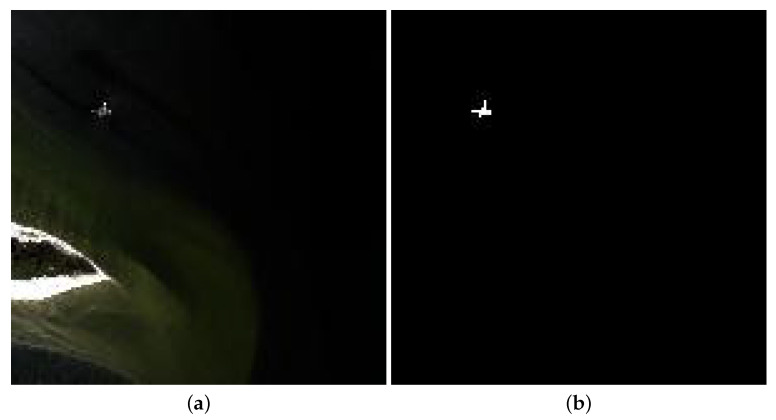
Beach dataset. (**a**) RGB composite image. (**b**) Ground truth.

**Figure 5 sensors-24-01652-f005:**
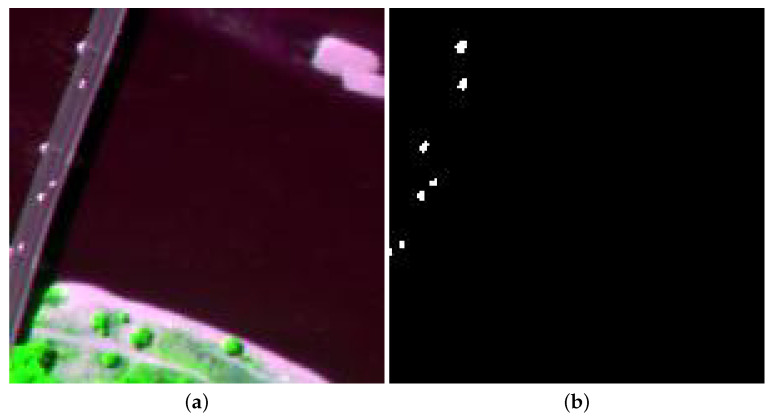
Pavia city dataset. (**a**) RGB composite image. (**b**) Ground truth.

**Figure 6 sensors-24-01652-f006:**
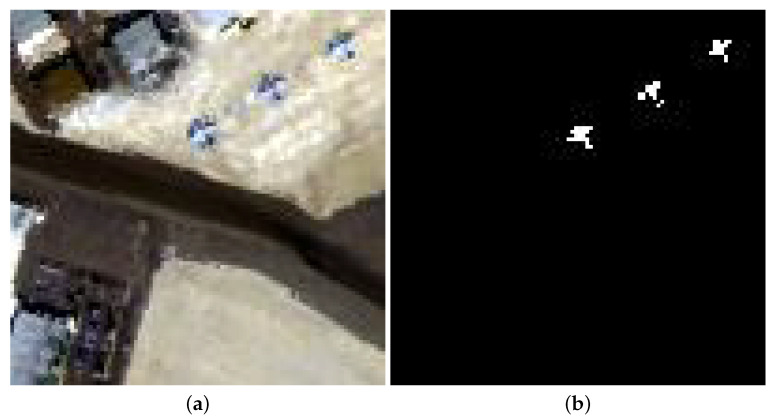
San Diego-I dataset. (**a**) RGB composite image. (**b**) Ground truth.

**Figure 7 sensors-24-01652-f007:**
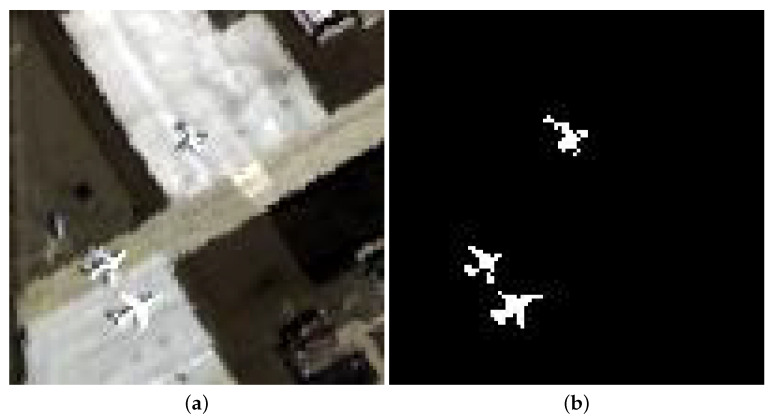
San Diego-II dataset. (**a**) RGB composite image. (**b**) Ground truth.

**Figure 8 sensors-24-01652-f008:**
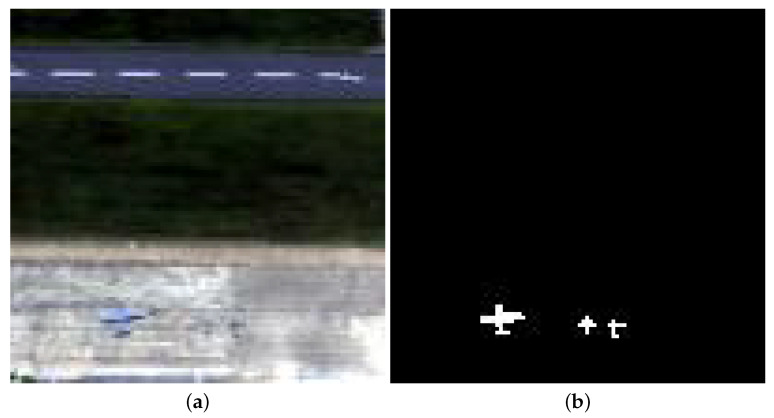
Gulfport dataset. (**a**) RGB composite image. (**b**) Ground truth.

**Figure 9 sensors-24-01652-f009:**
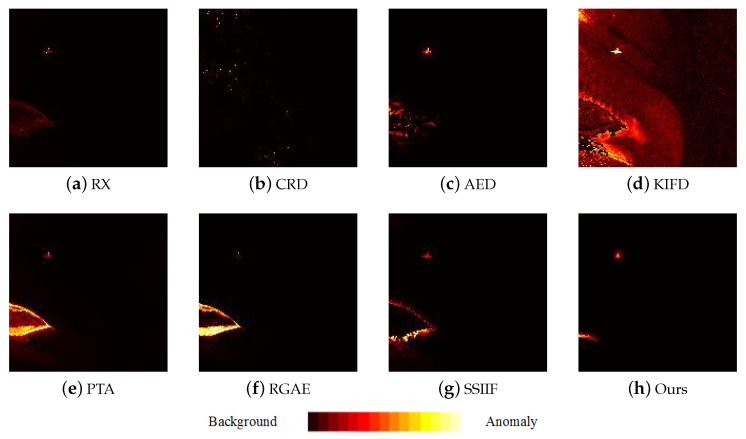
Detection maps of all approaches on Beach dataset. (**a**) RX. (**b**) CRD. (**c**) AED. (**d**) KIFD. (**e**) PTA. (**f**) RGAE. (**g**) SSIIF. (**h**) Ours.

**Figure 10 sensors-24-01652-f010:**
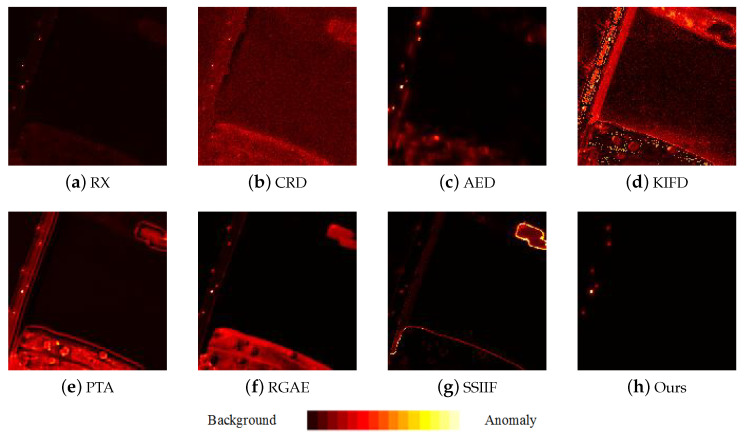
Detection maps of all approaches on Pavia city dataset. (**a**) RX. (**b**) CRD. (**c**) AED. (**d**) KIFD. (**e**) PTA. (**f**) RGAE. (**g**) SSIIF. (**h**) Ours.

**Figure 11 sensors-24-01652-f011:**
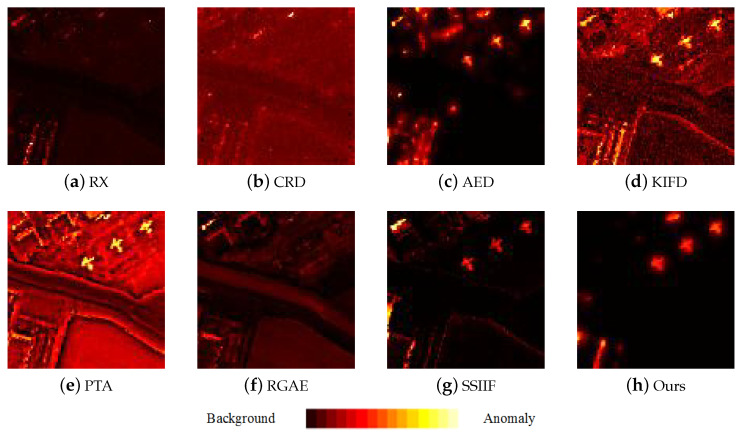
Detection maps of all approaches on San Diego-I dataset. (**a**) RX. (**b**) CRD. (**c**) AED. (**d**) KIFD. (**e**) PTA. (**f**) RGAE. (**g**) SSIIF. (**h**) Ours.

**Figure 12 sensors-24-01652-f012:**
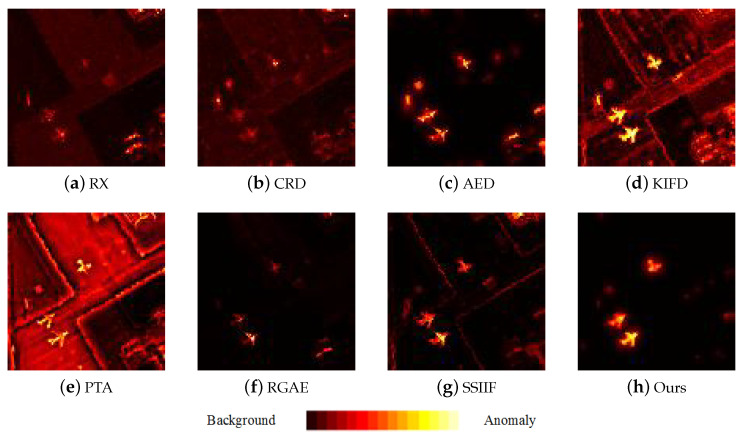
Detection maps of all approaches on San Diego-II dataset. (**a**) RX. (**b**) CRD. (**c**) AED. (**d**) KIFD. (**e**) PTA. (**f**) RGAE. (**g**) SSIIF. (**h**) Ours.

**Figure 13 sensors-24-01652-f013:**
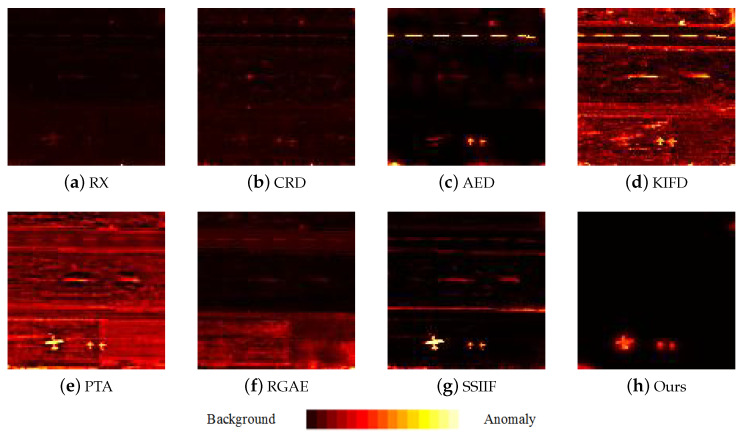
Detection maps of all approaches on Gulfport dataset. (**a**) RX. (**b**) CRD. (**c**) AED. (**d**) KIFD. (**e**) PTA. (**f**) RGAE. (**g**) SSIIF. (**h**) Ours.

**Figure 14 sensors-24-01652-f014:**
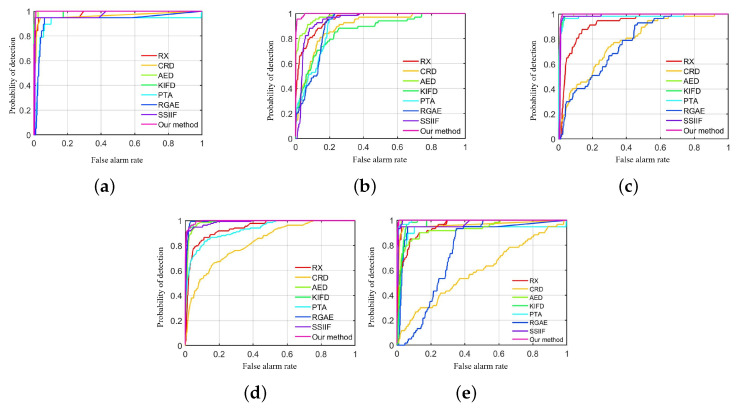
ROC curves of all detection techniques on all datasets. (**a**) Beach dataset, (**b**) Pavia city dataset. (**c**) San Diego-I dataset. (**d**) San Diego-II dataset. (**e**) Gulfport dataset.

**Figure 15 sensors-24-01652-f015:**
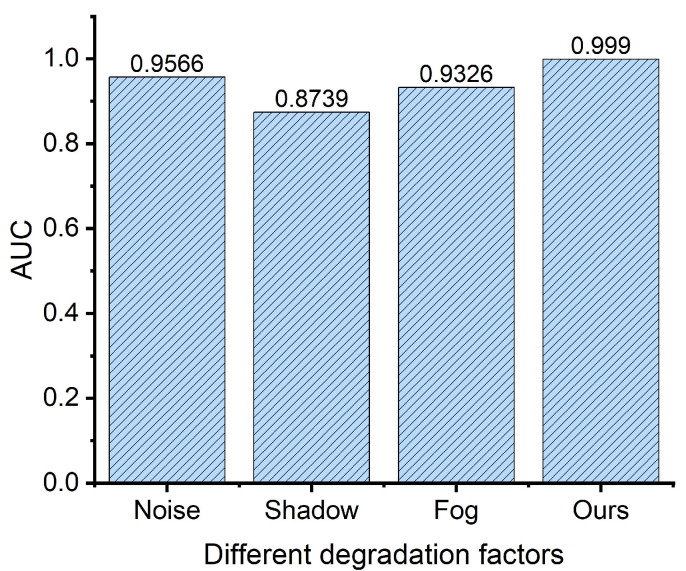
The influence of different degradation factors on anomaly detection.

**Figure 16 sensors-24-01652-f016:**
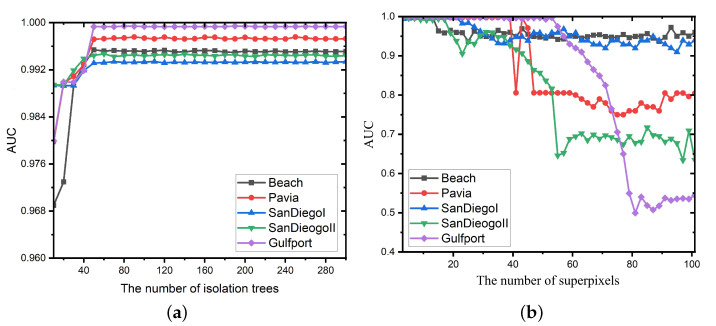
The influence of different parameters in the spectral anomaly detection branch. (**a**) Different numbers of isolation trees *q*. (**b**) Different numbers of superpixels *N*.

**Figure 17 sensors-24-01652-f017:**
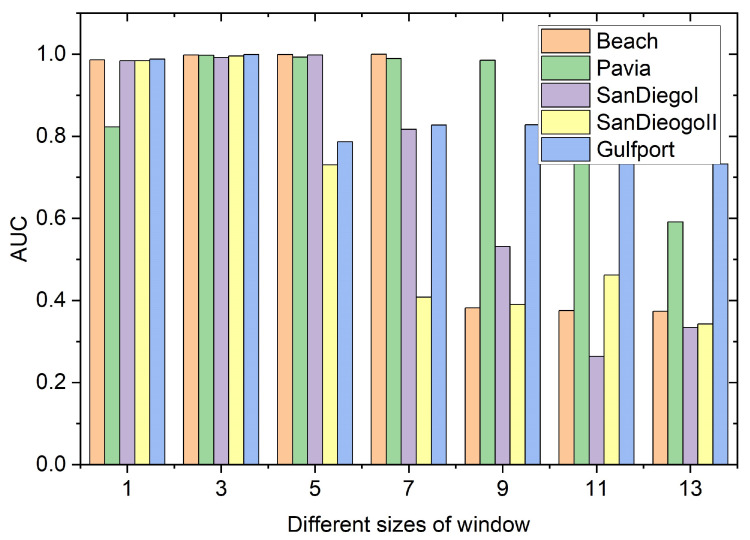
The influence of different sizes of window in the spatial anomaly detection branch.

**Table 1 sensors-24-01652-t001:** Detailed information of the used datasets.

Datasets	Beach	Pavia City	San Diego-I	San Diego-II	Gulfport
Sensor	AVIRIS	ROSIS	AVIRIS	AVIRIS	AVIRIS
Number of pixels	150 × 150	150 × 150	100 × 100	100 × 100	100 × 100
Spectral range	0.4–2.5	0.43–0.86	0.4–2.5	0.4–2.5	0.55–1.85
Number of spectral bands	188	205	189	126	191
Spatial resolution	17.2	1.3	3.5	3.5	3.4
Spectral resolution	10	15	10	10	10

**Table 2 sensors-24-01652-t002:** Detection results of all studied approaches. The best effect is displayed in bold.

Datasets	RX	CRD	AED	KIFD	PTA	RGAE	SSIIF	Ours
Beach	0.9807	0.9727	0.9974	0.9905	0.9184	0.9393	0.9672	**0.9978**
Pavia	0.9538	0.8941	0.9793	0.8742	0.9061	0.9042	0.9345	**0.9972**
San Diego-I	0.9219	0.7826	0.9915	0.9934	0.9791	0.7914	0.9775	**0.9949**
San Diego-II	0.9403	0.9687	0.9846	0.9931	0.9292	0.9929	0.9811	**0.9956**
Gulfport	0.9526	0.9618	0.9314	0.9683	0.9955	0.7583	0.9971	**0.9990**

**Table 3 sensors-24-01652-t003:** The detection effect of different components in the proposed framework.

Datasets	Beach	Pavia	San Diego-I	San Diego-II	Gulfport
Spatial branch	0.9866	0.9889	0.9844	0.9872	0.9967
Spectral branch	0.9790	0.9332	0.9833	0.9824	0.9918
Ours	0.9978	0.9972	0.9949	0.9956	0.9990

**Table 4 sensors-24-01652-t004:** Running time of different detection approaches.

Datasets	RX	CRD	AED	KIFD	PTA	RGAE	SSIIF	Ours
Beach	**0.24**	280.39	28.04	52.08	51.74	311.74	47.41	36.18
Pavia	**0.13**	274.38	31.64	61.41	31.09	181.79	33.64	31.73
San Diego-I	**0.13**	136.45	21.93	37.41	24.08	125.17	30.41	25.15
San Diego-II	**0.11**	142.62	19.92	32.73	24.33	147.51	28.33	24.51
Gulfport	**0.12**	119.01	21.65	37.31	24.57	140.86	29.73	28.26

## Data Availability

The data presented in this study are available on http://xudongkang.weebly.com/ (accessed on 27 December 2023).

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
