# Peer review of "Spectral–Spatial Feature Fusion for Hyperspectral Anomaly Detection"

_sensors, 2024, doi:10.3390/s24051652_

Round 1

Reviewer 1 Report

Comments and Suggestions for Authors

Comments on the Quality of English Language

Minor editing of English language required.

Reviewer 2 Report

Comments and Suggestions for Authors

The authors propose an advanced technique for hyperspectral anomaly detection. It is interesting topic but a few major concerns have to be addressed to consider this paper for publication in Sensors.

1. Please specify the task you try to solve. Are your results valid for any type HSI imager? Any objects (not only planes) and any backgrounds? If you insist on that, please add much more examples. Otherwise, please clearly limit the scope of your findings in Abstract and Conclusion.

2. Spectral mage quality a lot depends on signal-to-noise ratio (10.3390/rs9111166), optical design and aberrations (10.3390/ma14112984), vibrations (10.1016/S0030-3992(03)00084-7), weather conditions, stability of the relative position of the sensor and the object and many other factors.  Do the artefacts introduced by these factors decrease the efficiency of your algorithm? Please address these issues in Discussion with these and/or other relevant references.

3. What about radiometric, spectral and spatio-temporal calibration of the hyperspectral imaging system? It is a key problem if you speak about "spectral-spatial feature fusion". Here, you play with well-matched (ideal) datasets. What is necessary to do in order to apply your technique to real HSI data which is obviously very different from the ideal.

4. Please summarize your datasets in a table and describe them in detail: type of the imager, number of pixels, spectral range, number of spectral images, spatial and spectral resolution, exposure time, etc. The readers should clearly understand the scope of the presented numerical experiments.

5. AUC and ROC abbreviations are not introduced.

Comments on the Quality of English Language

Moderate English style and spell check is necessary.

Round 2

Reviewer 2 Report

Comments and Suggestions for Authors

The paper became clearer but I still miss the discussion on the scope of this method. My questions #2 and #3 are not reflected in the revised version. I think that these are key points for real experiments. The authors should at least address these issues and explain how they will influence the efficiency of the proposed technique and how to overcome them in practice.

2) Spectral image quality a lot depends on signal-to-noise ratio (10.3390/rs9111166), optical design and aberrations (10.3390/ma14112984), vibrations (10.1016/S0030-3992(03)00084-7), weather conditions, stability of the relative position of the sensor and the object and many other factors. Do the artefacts introduced by these factors decrease the efficiency of your algorithm? Please address these issues in Discussion with these and/or other relevant references.

3) What about radiometric, spectral and spatio-temporal calibration of the hyperspectral imaging system? It is a key problem if you speak about ”spectral-spatial feature fusion”. Here, you play with well-matched (ideal) datasets. What is necessary to do in order to apply your technique to real HSI data which is obviously very different from the ideal.

Comments on the Quality of English Language

Minor English style check is necessary.
